# Sociodemographic and behavioral factors associated with diet quality among low-income community health center patients with hypertension

Jessica Cheng[1,2]*, Katherine C. Faulkner[2], Ashlie Malone[2], Kristine D. Gu[3,4], Anne N. Thorndike[2,3]

1 Department of Epidemiology, Harvard T H Chan School of Public Health, Boston, Massachusetts, United States of America, 2 Division of General Internal Medicine, Massachusetts General Hospital, Boston, Massachusetts, United States of America, 3 Harvard Medical School, Boston, Massachusetts, United States of America, 4 Division of Endocrinology, Massachusetts General Hospital, Boston, Massachusetts, United States of America

* jcheng28@mgh.harvard.edu

**Data Availability Statement:** All relevant data for this study are publicly available from the OSF repository (https://doi.org/10.17605/OSF.IO/WY4XF).

## Abstract

### Objective

Identify the most important sociodemographic and behavioral factors related to the diet of low-income adults with hypertension in order to guide the development of a community health worker (CHW) healthy eating intervention for low-income populations with hypertension.

### Design

In this cross-sectional analysis, dietary recalls were used to assess Healthy Eating Index-2020 (HEI-2020) total (range: 0 to 100 [best diet quality]) and component scores and sodium intake. Self-reported sociodemographic and behavioral data were entered into a Least Absolute Shrinkage and Selection Operator (LASSO) regression model to determine the relative importance of factors related to diet quality.

### Setting

Five community health centers in Boston, Massachusetts.

### Participants

Adults (>20 years old) with a hypertension diagnosis.

### Results

Participants (N = 291) were mostly female (65.0%), on Medicaid (82.8%), food insecure (59.5%), and Hispanic (52.2%). The mean (95% CI) HEI-2020 score was 63.0 (62.3, 65.7) Component scores were low for sodium and whole grains; mean (SE) sodium intake was 2676.9 (45.5) mg/day. The most important factors associated with lower HEI-2020 scores

**Funding:** Research reported in this publication was supported by the National Heart, Lung, and Blood Institute of the National Institutes of Health under Award Number T32HL098048 (Cheng) and K24 HL163073 (Thorndike) and the National Institute of Diabetes and Digestive and Kidney Diseases under Award Number R01 DK124145 (Thorndike) and T32DK007028 (Gu). The content is solely the responsibility of the authors and does not necessarily represent the official views of the National Institutes of Health. The funders had no role in study design, data collection and analysis, decision to publish, or preparation of the manuscript.

**Competing interests:** The authors have declared that no competing interests exist.

were: not having own housing, male gender, tobacco use, marijuana use, and skipping meals; the most important factors associated with higher HEI-2020 scores were Hispanic ethnicity and receipt of community food resources (5-fold cross-validated $R^2 = 0.17$).

## Conclusions

In this population of low-income adults with hypertension, diet quality would be improved by reducing sodium and increasing whole grain intake. Healthy eating interventions among low-income populations should consider providing dietary guidance in the context of behavioral factors (e.g., meal skipping) and substance use (e.g., marijuana) and should address barriers to health eating through referral to community food resources (e.g., food pantries).

## Introduction

Hypertension, a leading risk factor for cardiovascular disease, affects nearly half of US adults [1, 2]. Healthy dietary patterns (e.g., the dietary approaches to stop hypertension [DASH] diet) that emphasize reducing sodium and saturated fat intake and adequate potassium-rich fruits and vegetables and whole grains have been shown to effectively reduce blood pressure [3] and 10-year atherosclerotic cardiovascular disease risk [4]. As such, referral to behavioral counseling interventions that address diet is recommended by the US Preventive Services Task Force (USPSTF) for patients with hypertension to reduce CVD risk [5].

However, low-income individuals may face unique challenges to achieving a healthy diet such as cost-related issues that prevent them from obtaining nutritious foods, [6] as well as barriers to food preparation including lack of time, cooking tools and appliances, and adequate storage [7]. Thus, recognizing that existing behavioral interventions may insufficiently address the context in which dietary decisions are made among under-resourced populations, the USPSTF specifically calls for more research on behavioral intervention approaches that work in low-resource settings [5].

Community health workers (CHWs) are lay individuals trained and certified in a particular area of chronic disease management education who are often members of the communities they serve. The American Heart Association suggests that, embedded in care teams, CHWs may be able to provide evidence-based nutritional advice in the context of the sociodemographic, behavioral, and cultural factors that may be barriers to adopting healthier dietary patterns among low-resourced communities [8, 9].

The purpose of this analysis was to 1) describe components of the current dietary pattern most in need of improvement among low-income adults with hypertension and 2) determine the most important sociodemographic and behavioral factors associated with their diet quality. The most important factors related to diet quality in this population will be used to guide the development and delivery of a CHW healthy eating intervention to improve diet quality for low-income adults with hypertension (ClinicalTrials.gov identifier: NCT06358417).

## Methods

This is a secondary analysis of the LiveWell observational cohort of community health center patients. LiveWell was designed to assess, in a natural experiment, the impacts of the Massachusetts Medicaid Flexible Services program that provides funding to accountable care organizations (ACOs) to connect patients with food and housing resources [10]. LiveWell

participants were recruited from five community health centers in the Boston, Massachusetts area. To be eligible, participants had to be between ages 21 to 62 years, have had at least two primary care visits in the previous two years, speak English or Spanish, and have Medicaid or commercial ACO insurance. This secondary analysis utilized baseline data collected from December 19, 2019 to December 8, 2020 among LiveWell participants with a hypertension diagnosis, determined by International Classification of Diseases (ICD)-10 code or by inclusion of hypertension in the problem list. Data were analyzed from May 12, 2023 to August 4, 2023. This study was conducted according to the guidelines laid down in the Declaration of Helsinki and all procedures involving research study participants were approved by the Mass General Brigham Institutional Review Board. Written informed consent was obtained from all subjects/patients.

The purpose of this analysis was to explore the relative importance of sociodemographic and behavioral factors to a dietary pattern that impacts blood pressure control. Therefore, all relevant factors obtained through survey, dietary recall, and medical record were included in modeling. For categorical variables with three or more levels, each level was entered into the model. A total of 72 factors were considered and are described below.

## Sociodemographic and behavioral factors

All sociodemographic and behavioral factors were self-reported in a survey. Demographics included age, gender, race, ethnicity, marital status, household size, number of children in the home, household income, education, and employment status. Gender, race, education, marital status, and employment categories were collapsed because of small cell sizes. Insurance status was determined from the medical record.

Food security was assessed using the 10-item US Department of Agriculture Adult Food Security Survey Module [11]. Using the Nutrition Environment Measures Survey on perceived nutrition environment, [12] participants were coded as having or not having 1) a freezer (attached or stand-alone), 2) other countertop appliances (e.g., toaster oven), and 3) other key appliances (i.e., refrigerator, microwave oven, stove, and oven). The average number of foods reported and whether main meals (i.e., breakfast, lunch, and dinner) were skipped were ascertained from dietary recalls. Whether the supermarket or another outlet was the primary shopping location was self-reported.

Participants self-reported receipt of food assistance from: 1) friends/family, 2) meals eaten at community organizations (e.g., soup kitchen), 3) meals delivered to the home (e.g., Meals on Wheels), 4) food from a community organization (e.g., food pantry), 5) supermarket gift cards, 6) Supplemental Nutrition Assistance Program, or 7) other programs (e.g., food from health clinic, Special Supplemental Nutrition Program for Women, Infants, and Children [WIC]).

Participants were considered to have housing instability if they answered yes to at least one question on a 3-item scale [13, 14] including not renting or owning a house or apartment, moving two or more times in the past twelve months, and worrying about not having housing in the next two months. Each housing instability item was also included individually in the model as a binary factor. Participants were categorized as having living site problems if they endorsed any of nine household problems (e.g., presence of pests). Participants were considered to have received transportation assistance if they endorsed receiving any of six types of assistance (e.g., reduced public transportation fares) and housing assistance if they endorsed receiving any of seven types of assistance (e.g., housing vouchers). Participants were considered to have received help obtaining food, housing, or transportation assistance if they endorsed help from any of six types of individuals (e.g., social worker) (S1 File).

Anxiety symptoms were assessed using the 7-item version of the General Anxiety Disorders Questionnaire [15]. Perceived stress was measured with the 10-item Perceived Stress Scale [16]. Depression symptoms were measured with the 8-item Patient Health Questionnaire [17]. Financial stress was measured using a 5-item subscale from the Weekly Stress Inventory [18]. Items adapted from the National Health Interview Survey were used to assess the delay of medical care due to financial reasons [19]. The 8-item modified Medical Outcomes Study Social Support Survey [20] measured emotional/informational, tangible, affectionate, and positive social interaction support. Participants additionally self-reported how often they felt lonely or isolated from those around them on a 5-point Likert scale. Participants' experience with discrimination was self-reported with the validated Everyday Discrimination Scale (Short Version) [21]. These measures were input as continuous variables.

Participants self-reported having difficulty with everyday functions because of vision, cognition, mobility, self-care (i.e., dressing or bathing), and independent living (i.e., running errands) impairments using wording from the Behavioral Risk Factor Surveillance System [22]. Weekly minutes spent sitting, walking, and engaging in moderate and vigorous physical activity were measured using the International Physical Activity Questionnaire short form [23]. Sleep quality was measured using the Patient-Reported Outcomes Measurement Information System (PROMIS) Sleep Disturbance short form 4a [24]. Participants self-reported daily cigarette use, binge drinking, cannabis/marijuana use, and prescription drug use for non-medical reasons/illegal drug use.

## Blood pressure

In this analysis, we used information extracted from the electronic health record in the Enterprise Data Warehouse to categorize blood pressure control, expanding upon the work from our previous publication [25] where uncontrolled blood pressure was determined via research patient data registry records only. Hypertension was considered uncontrolled if mean systolic blood pressure $\geq$ 140 mm Hg or mean diastolic blood pressure $\geq$ 90 mm Hg in the 12 months prior to baseline. Participants with no blood pressure recorded in the previous 12 months were considered to have unknown hypertension status.

## Dietary assessment

Two dietary recalls were collected at baseline using the National Cancer Institute Automated Self-Administered 24-Hour Recall System (ASA24-2020), [26] which has shown acceptable accuracy [27] compared to other dietary assessment methods. Dietary recalls were used to estimate sodium intake, potassium intake, and diet quality.

We chose to use the Healthy Eating Index-2020 (HEI-2020) [28, 29] as the measure of diet quality in this study as it is a valid and reliable measure of overall diet quality aligned with the Dietary Guidelines for Americans. The HEI has been shown to relate to blood pressure, [30, 31] correlate with other diet quality indices, and shows a similar relationship to CVD mortality as other indices [32]. The HEI-2020 total score is a summation of 9 adequacy (i.e., total fruit, whole fruit, total vegetables, beans and greens, total protein, seafood and plant protein, whole grains, dairy, and fatty acid ratio) and 4 moderation (i.e., saturated fat, added sugar, sodium, refined grains) subcomponent scores. The minimum score for all subcomponents is 0, representing the worst intake for that component. Subcomponents scores are based on intake to calories for all components except for the "Fatty Acid Ratio" score, which is based on the ratio of unsaturated fatty acids to saturated fatty acids and "Added Sugars" and "Saturated Fats" scores which are based on percent of energy intake. The total HEI-2020 score ranges from 0 to 100

with 100 indicating the best possible diet quality through maximization of all subcomponents scores.

## Analysis plan

HEI-2020 total and component scores were calculated using the population ratio method [33]. Bootstrap resampling with 200 bootstraps was utilized to estimate 95% confidence intervals (CIs) using the percentile method as appropriate since the bootstrap distribution was approximately normal. Radar plots were used to visualize HEI-2020 scores for patients with hypertension.

The National Cancer Institute's Simulating Intake of Micronutrients for Policy Learning and Engagement (SIMPLE) macro was used to describe the mean and distribution of sodium and potassium intake from dietary recalls [34]. A detailed explanation of the use of this macro along with information on the availability of SAS code is available elsewhere [34]. Estimates were adjusted for language of administration (i.e., English or Spanish), mode of administration (i.e., staff-administered by phone vs. self-administered online), whether recalls were consecutive, a weekend/weekday indicator, and participant rating of whether the recall represented usual intake. An intake level of 1500 mg/day, the ideal limit for sodium endorsed by the American Heart Association (AHA), was used [35]. The percent whose intake is above the AHA level represents the proportion who could benefit from reducing sodium intake. There is no estimated average requirement (EAR) value for potassium intake; therefore, we examined the proportion above sex-specific adequate intake (AI) values of 2400 and 2600 mg/day for women and men, respectively [36]. The percent above this cut point is likely to consume adequate potassium as it represents the proportion with intake above that of apparently healthy individuals. Analyses were performed using Statistical Analysis Systems statistical software package version 9.4 (SAS Institute, Cary, NC, USA).

To identify factors related to diet quality, the simple scoring algorithm was used to calculate individual level estimates of HEI-2020 total scores [33]. Single imputation was used for missing values of factors using *PROC MI*. Most factors had less than 10 observations missing (S1 Table). More than 10 observations were missing for having a freezer, having other kitchen appliances, and for physical activity; however, missingness was less than 20% for these factors.

Least absolute shrinkage and selection operator (LASSO) regression was used to identify the most important factors related to HEI-2020 total scores. Factors were standardized such that the absolute value of the coefficients could be used to determine importance. K-fold (k = 5) cross validation was used to select the best model. In sensitivity analyses, elastic net with $\alpha = 0.5$ and 5-fold cross validation was also used. Analyses were performed in R using package *glmnet* [37]. In exploratory analyses, HEI-2020 total and component scores and radar plots were also assessed for the top four factors identified in the LASSO model.

## Results

The sample (N = 291) was mostly female (65.0%) with a median (p25, p75) age of 54.0 (47.0, 58.0) years (Table 1). Over half of participants were food insecure (59.5%), identified as Hispanic (52.2%), and had Medicaid insurance (82.8%). Most participants' blood pressure was controlled (76.6%), with mean blood pressure less than 140/90 mmHg.

The mean (95% CI) HEI-2020 total score was 63.0 (62.3, 65.7) points (Table 2 and S1 Fig). Total protein, seafood and plant protein, whole fruit, and greens and beans were the subcomponents with the highest scores. Sodium and whole grains were the subcomponents with the lowest scores. The mean (SE) sodium intake was 2676.9 (45.5) mg, and based on this result, approximately 86.6% (1.6%) would be estimated to have excessive sodium intake above 1500

**Table 1. Characteristics of a sample of low-income community health center patients with health insurance and a hypertension diagnosis (N = 291).**

| | | Median (p25, p75)/ N (%) |
|---|---|---|
| Age | | 54.0 (47.0, 58.0) |
| Female sex | | 189 (65.0%) |
| Ethnicity | | 152 (52.2%) |
| Race | White, only | 141 (48.5%) |
| | Back, only | 54 (18.6%) |
| | Other races, only* | 62 (21.3%) |
| | More than One Race | 34 (11.7%) |
| Insurance, Medicaid | | 241 (82.8%) |
| Married/Living with Significant Other | | 109 (37.5%) |
| Employment | Full-time | 57 (19.6%) |
| | Part-time† | 51 (17.5%) |
| | Homemaker/work in the home | 30 (10.3%) |
| | Not employed, Looking for work§ | 35 (12.0%) |
| | Not employed, Not looking for work | 34 (11.7%) |
| | Disabled and unable to work | 84 (28.9%) |
| Education | Less than high school | 66 (22.7%) |
| | High School | 87 (29.9%) |
| | Some College | 84 (28.9%) |
| | College or more | 54 (18.6%) |
| US Department of Agriculture Food Security raw score‖ | | 3.0 (1.0, 7.0) |
| Housing Security | | 120 (41.2%) |
| Insurance Type, Medicaid | | 241 (82.8%) |
| Household income <$30k | | 206 (70.8%) |
| Supplemental nutrition assistance program (SNAP) | | 175 (60.1%) |
| Current smoking | | 83 (28.5%) |
| Hypertension status¶ | | |
| Controlled | | 223 (76.6%) |
| Uncontrolled | | 47 (16.2%) |
| Unknown control | | 21 (7.2%) |

*Other race includes Asian only, Native Hawaiian/Pacific Islander only, Native American/Native Alaskan only, and Other race only.

†Includes one student who also works part-time.

§Includes those who are retired.

‖High food security: Raw score 0; Marginal food security: Raw score 1–2; Low food security: Raw score: 3–5; Very low food security Raw score 6–10

¶Participants in the unknown HTN control group had 0 day with a BP measurement in the year before enrollment. If 2 or more BP readings in 1 day, only the lowest BP was included.

mg/day. The mean (SE) potassium intake of female and male participants with hypertension was 2137.8 (31.8) and 2177 (35.2) mg, indicating that 32.5% (1.6%) and 26.1% (1.6%) would be estimated to consume adequate potassium, respectively.

In order of importance, LASSO regression identified the top factors related to HEI-2020 total scores (Fig 1). The factors associated with lower diet quality were not having own housing, male gender, tobacco use, marijuana use, and skipping meals, and the factors associated with higher diet quality were Hispanic ethnicity and receipt of community food resources (e.g., eating meals at a community organization and receiving a supermarket gift card). The

**Table 2. Healthy eating index-2020 total and subcomponent scores and bootstrapped 95% confidence intervals.**

| | Max Score | Total Sample (N = 291) | Does Not Have Own Housing (N = 42) | Has Own Housing (N = 244) | Female (N = 189) | Male (N = 102) | Hispanic (N = 152) | Non-Hispanic (N = 139) | Smoking (N = 81) | Non-Smoking (N = 207) |
|---|---|---|---|---|---|---|---|---|---|---|
| Total Score | 100 | 63.0 (62.3, 65.7) | 53.2 (47.0, 59.1) | 65.5 (62.7, 67.7) | 67.5 (65.1, 69.9) | 58.6 (53.5, 62.2) | 67.5 (65.0, 70.0) | 58.5 (54.7, 61.9) | 55.9 (50.0, 60.3) | 66.2 (63.7, 68.3) |
| **Adequacy Components** | | | | | | | | | | |
| Total Vegetables | 5 | 4.1 (4.2, 4.8) | 3.7 (3.1, 4.6) | 4.6 (4.2, 5.0) | 4.9 (4.4, 5.0) | 3.9 (3.4, 4.3) | 5.0 (4.9, 5.0) | 3.7 (3.2, 4.1) | 3.7 (3.2, 4.3) | 4.8 (4.2, 5.0) |
| Greens and Beans | 5 | 5.0 (4.8, 5.0) | 3.5 (1.8, 5.0) | 5.0 (4.9, 5.0) | 5.0 (5.0, 5.0) | 4.5 (3.4, 5.0) | 5.0 (5.0, 5.0) | 3.9 (3.0, 5.0) | 3.9 (2.5, 5.0) | 5.0 (5.0, 5.0) |
| Total Fruit | 5 | 3.5 (3.3, 4.1) | 2.7 (1.7, 3.8) | 3.7 (3.2, 4.3) | 4.1 (3.5, 4.9) | 3.0 (2.4, 3.8) | 4.6 (3.9, 5.0) | 2.8 (2.2, 3.3) | 2.4 (1.7, 3.1) | 4.1 (3.6, 4.7) |
| Whole Fruit | 5 | 4.8 (4.2, 5.0) | 3.1 (1.9, 4.4) | 5.0 (4.2, 5.0) | 5.0 (4.7, 5.0) | 3.7 (2.6, 4.9) | 5.0 (5.0, 5.0) | 3.6 (2.7, 4.5) | 2.7 (1.6, 4.0) | 5.0 (4.8, 5.0) |
| Whole Grains | 10 | 2.8 (2.4, 3.2) | 1.5 (0.8, 2.3) | 3.0 (2.6, 3.5) | 3.2 (2.7, 3.8) | 2.3 (1.7, 3.1) | 2.7 (2.0, 3.2) | 2.9 (2.3, 3.5) | 2.5 (1.9, 3.3) | 3.0 (2.5, 3.5) |
| Dairy | 10 | 5.8 (5.8, 6.8) | 6.5 (5.2, 8.2) | 6.1 (5.5, 6.8) | 5.9 (5.1, 6.7) | 6.7 (6.0, 7.6) | 5.5 (4.8, 6.4) | 6.9 (6.2, 7.6) | 6.6 (5.5, 7.7) | 6.2 (5.6, 6.8) |
| Total Protein Food | 5 | 5.0 (5.0, 5.0) | 5.0 (5.0, 5.0) | 5.0 (5.0, 5.0) | 5.0 (65.0, 5.0) | 5.0 (5.0, 5.0) | 5.0 (5.0, 5.0) | 5.0 (5.0, 5.0) | 5.0 (5.0, 5.0) | 5.0 (5.0, 5.0) |
| Seafood and Plant Protein | 5 | 5.0 (5.0, 5.0) | 4.8 (2.8, 5.0) | 5.0 (5.0, 5.0) | 5.0 (5.0, 5.0) | 5.0 (4.0, 5.0) | 5.0 (5.0, 5.0) | 4.9 (4.0, 5.0) | 5.0 (4.1, 5.0) | 5.0 (5.0, 5.0) |
| Fatty Acid Ratio | 10 | 4.6 (4.1, 5.1) | 4.2 (3.0, 5.2) | 4.7 (4.1, 5.4) | 5.0 (4.3, 6.1) | 4.2 (3.4, 5.0) | 5.2 (4.3, 6.1) | 4.1 (3.5, 4.8) | 3.8 (2.9, 4.7) | 4.9 (4.3, 5.7) |
| **Moderation Components** | | | | | | | | | | |
| Sodium | 10 | 2.2 (1.4, 2.5) | 0.0 (0.0, 1.8) | 2.3 (1.6, 3.1) | 2.5 (1.8, 3.3) | 1.2 (0.1, 2.2) | 1.8 (0.9, 2.8) | 2.1 (1.2, 3.0) | 2.3 (1.3, 3.2) | 1.9 (1.1, 2.6) |
| Refined Grains | 10 | 6.6 (6.1, 7.2) | 5.2 (3.7, 6.8) | 7.0 (6.4, 7.6) | 7.4 (6.8, 8.0) | 5.8 (4.7, 6.7) | 6.9 (6.1, 7.8) | 6.5 (5.7, 7.2) | 6.4 (5.3, 7.6) | 6.8 (6.0, 7.4) |
| Saturated Fat | 10 | 5.7 (5.3, 6.3) | 5.1 (3.9, 6.3) | 5.9 (5.3, 6.5) | 6.4 (5.8, 7.1) | 4.9 (4.0, 5.9) | 6.7 (6.0, 7.5) | 4.9 (4.2, 5.7) | 4.6 (3.4, 5.6) | 6.2 (5.5, 6.8) |
| Added Sugar | 10 | 7.9 (7.6, 8.4) | 7.8 (6.6, 8.9) | 8.1 (7.5, 8.7) | 8.0 (7.3, 8.6) | 8.4 (7.3, 9.2) | 9.1 (8.3, 9.8) | 7.2 (6.6, 7.9) | 7.2 (6.2, 8.1) | 8.4 (7.7, 9.0) |

Note: The minimum score for all subcomponents is 0. Subcomponent scores are based off intake to calories for all components except for the "Fatty Acid Ratio" score which is based off the ratio of unsaturated fatty acids to saturated fatty acids and "Added Sugars" and "Saturated Fats" scores which are based on percent of energy intake

cross-validated $R^2$ was 0.17. A sensitivity analysis using elastic net ranked factors similarly with a cross-validated $R^2$ of 0.18 (S2 Fig).

In exploratory analyses of the top four factors associated with diet quality, the mean (95% CI) HEI-2020 total scores of individuals who do not have their own housing (53.2 [47.0, 59.1]), non-Hispanic participants (58.5 [54.7, 61.9]), men (58.6 [53.5, 62.2]), and individuals who smoke (55.9 [50.0, 60.3]) were about 10 points lower compared to those with their own housing (65.5 [62.7, 67.7]), Hispanic participants (67.5 [65.0, 70.0]), women (67.5 [65.1, 69.9]), and non-smokers (66.2 [63.7, 68.3]), respectively. (Table 2 and S3 Fig).

## Discussion

This study of low-income community health center patients with hypertension demonstrated that overall diet quality could be improved to better align with dietary guidelines [38]. Similar to the overall U.S. population, [39] the consumption of sodium was high and the consumption of whole grains was low given caloric intake. Furthermore, the results also suggested that over

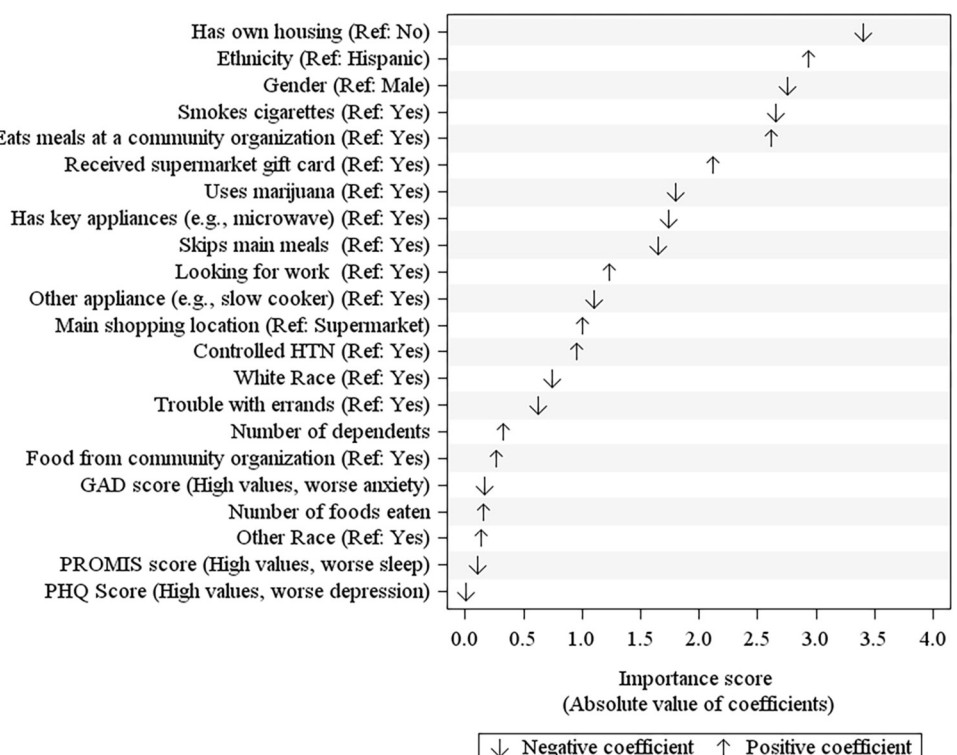

**Fig 1. Factors related to HEI-2020 scores identified by LASSO.** Note: GAD, General Anxiety Disorders Questionnaire; PHQ, Patient Health Questionnaire; PROMIS, Patient-Reported Outcomes Measurement Information System. Factors are ranked from top to bottom in order of importance based on the absolute value of the model coefficients. A positive coefficient means that the reference value is associated with better diet quality. A negative coefficient means that the reference value is associated with worse diet quality.

half of patients could benefit from increasing potassium consumption. Lower sodium intake and higher potassium intake have been associated with lower risk of cardiovascular disease [40]. To address diet quality in this at-risk population, the adoption of multiple dietary patterns such as DASH, the Mediterranean diet, or a vegetarian diet may be recommended in line with personal and cultural preferences [41, 42]. However, adoption and adherence to these dietary patterns can be difficult, especially among low-resourced groups. This analysis identified important sociodemographic and behavioral factors associated with diet among community health center patients with hypertension that can guide the development of a CHW healthy eating intervention. We caution that the associations between factors and diet quality cannot be interpreted causally; therefore, addressing any individual factor may not necessarily lead to direct improvement in diet quality.

Results suggest that gender and ethnicity were among the most important factors related to diet quality. While women and Hispanic individuals had diet quality scores about 10 points higher than their counterparts, all subgroups had overall diet quality below the score of 74 that would meet Healthy People 2020 goals [43]. Interventions to address diet may need to consider tailoring advice and support to account for differences by gender and culture.

CHW-delivered hypertension programs are promising ways of reaching diverse participants in consultation with, and outside of, primary care [44, 45]. Other CHW-led interventions with dietary components show potential effectiveness in reducing blood pressure and improving health behaviors although diet quality measures, such as the HEI, are not evaluated [46–48]. As with other CHW-led hypertension interventions, our planned intervention

development and evaluation will include qualitative work to complement our quantitative analyses [46, 49]. Thus, we have planned semi-structured interviews with both patients and CHWs, in order to contextualize our current findings and to explore if and how to address these factors in our intervention design and delivery (e.g., involving female partners in the intervention). Thus, our work utilizes similar practices as other promising interventions and expands on existing literature as we will explicitly focus on robust measurement of diet quality outcomes.

In the current study, housing situation was the top factor related to diet quality. Prior research has demonstrated that unstable housing is associated with poor cardiometabolic health [50]. Therefore, programs that support finding housing and applying for rental assistance programs for those in need would enhance efforts to improve diet and hypertension. Similarly, connecting patients with community-based nutrition services may help improve the diet of individuals with hypertension as our model suggested that receiving support for food acquisition (i.e., meals at a community organization and receiving a supermarket gift card) was positively associated with diet quality in this mostly low-income population. The role of CHWs includes assessment of social needs and connection with social services, advocacy, and support [51]. This uniquely situates them to being able to address housing and food barriers. Our intervention may thus include referral to community resources.

Our study identified that the use of cigarettes and marijuana was associated with lower diet quality. In the US population, smoking has been associated with lower diet quality [52] and quitting smoking may be associated with improvement in diet and other health metrics (e.g., mental health) [53]. Research has also demonstrated that cannabis users have lower diet quality compared to never users [54]; however, the reasons for this lower diet quality are unclear. While there may be no causal link between these behaviors and diet quality, CHWs might address concerns around weight gain when quitting smoking if such concerns arise [55].

The current study also demonstrated that skipping meals was associated with lower diet quality. The effect of meal skipping is complex with one analysis of the National Health and Nutrition Examination Survey data suggesting that meal skipping is generally associated with lower daily caloric intake and lower diet quality, but also improvements in the consumption of certain dietary components (e.g., reduced sodium intake when skipping lunch or dinner) [56]. Understanding an individual's reasons for meal skipping would be important for providing effective dietary advice for individuals with hypertension. For example, a CHW could address reasons such as unaffordability of healthy food or not having time to prepare meals with strategies like grocery budgeting, meal planning, and quick healthy go-to meals. We note that while participation in a phone- and web-based CHW-led program is a temporary time constraint while enrolled, we hope that CHWs can help patients develop and/or strengthen skills that reduce the time burden associated with shopping and cooking.

## Strengths and limitations

A strength of this analysis is the use of estimation methods that account for the substantial within-person variability in dietary intake to leverage dietary recalls for estimating usual dietary intake. This formative research identifies important sociodemographic and behavioral factors related to diet quality that will complement future stakeholder interviews to provide a more in-depth understanding of these facilitators and barriers to adopting a healthier dietary pattern. This will ultimately inform the development and delivery of a CHW-led dietary intervention for low-income adults with hypertension.

A limitation is that difficulties using the ASA-24 search function to find foods and uncertainty as to how to proceed through the recall have been reported among low-income

individuals which may reduce the accuracy of dietary measures [57]. However, participants in this study were given the option to complete recalls with study staff by phone, if desired. Additionally, factors identified in the LASSO analysis may not be causally related to HEI-2020 total scores; therefore, additional research is needed to understand associations between identified factors and diet quality. Finally, this was a relatively small sample of patients from one urban health system in the northeast US and data collection occurred during the COVID-19 pandemic. Thus, results may not be generalizable to patients from rural areas or from other regions within the US or outside the US, and the pandemic might have temporarily increased the reporting of social needs and/or lead to the worsening of diet quality.

## Conclusion

Among low-income adults with hypertension, sociodemographic and behavioral factors were associated with diet quality. Participants had high intake of sodium, low intake of potassium, and low intake of whole grains in the context of their caloric intake. Social factors, such as housing and community-based nutrition assistance, and behavioral factors, such as meal skipping, were important factors associated with diet quality. These results support AHA recommendations for using community health workers as part of team-based care to improve implementation of blood pressure control strategies, with a particular focus on lifestyle modification [9]. In the future, CHWs could be trained to provide evidence-based dietary advice to improve blood pressure management and reduce cardiovascular risk in the context of specific sociodemographic and behavioral barriers and facilitators of diet quality in low-income populations.

## Supporting information

**S1 Fig. Healthy eating index-2020 total score among patients with hypertension.** Note: Scores touching the outer ring represent the maximum score for a subcomponent (100% of the maximum score). A perfect diet quality score of 100 would be represented by touching the outer ring for all subcomponents.
(TIF)

**S2 Fig. Factors related to healthy eating index-2020 scores identified by elastic net.** Note: GAD, General Anxiety Disorders Questionnaire; PHQ, Patient Health Questionnaire; PROMIS, Patient-Reported Outcomes Measurement Information System Sleep Disturbance Questionnaire. Factors are ranked from top to bottom in order of importance based on the absolute value of the model coefficients. A positive coefficient means that the reference value is associated with better diet quality. A negative coefficient means that the reference value is associated with worse diet quality.
(TIF)

**S3 Fig. Healthy eating index-2020 total scores among patients with hypertension by select factors.** Note: A = HEI-2020 Total Scores by Housing Situation; B = HEI-2020 Total Scores by Ethnicity; C = HEI-2020 Total Scores by Gender; D = HEI-2020 Total Scores by Smoking; Total Scores by ethnicity. Scores touching the outer ring represent the maximum score for a subcomponent (100% of the maximum score). A perfect diet quality score of 100 would be represented by touching the outer ring for all subcomponents.
(TIF)

**S1 Table. Number of participants with missing data for factors including in the LASSO model.** Note: There were 0 participants with missing values for all other variables included in

modeling.
(DOCX)

**S1 File. Supplemental information on food, housing, and transportation insecurity questions.**
(DOCX)

## Author Contributions

**Conceptualization:** Jessica Cheng, Anne N. Thorndike.

**Formal analysis:** Jessica Cheng.

**Funding acquisition:** Anne N. Thorndike.

**Methodology:** Jessica Cheng.

**Writing – original draft:** Jessica Cheng.

**Writing – review & editing:** Katherine C. Faulkner, Ashlie Malone, Kristine D. Gu, Anne N. Thorndike.

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
