## [Decision Letter · Decision Letter 0]

1 Jul 2024

PONE-D-24-06116Sociodemographic and Behavioral Factors Associated with Diet Quality among Low-income Community Health Center Patients with HypertensionPLOS ONE

Dear Dr. Cheng,

Thank you for submitting your manuscript to PLOS ONE. After careful consideration, we feel that it has merit but does not fully meet PLOS ONE’s publication criteria as it currently stands. Therefore, we invite you to submit a revised version of the manuscript that addresses the points raised during the review process.

We look forward to receiving your revised manuscript.

Kind regards,

Habiba I. Ali, PhD, RD, CDE

Academic Editor

PLOS ONE

“Research reported in this publication was supported by the National Heart, Lung, and Blood Institute of the National Institutes of Health under Award Number T32HL098048 (Cheng) and K24 HL163073 (Thorndike) and the National Institute of Diabetes and Digestive and Kidney Diseases under Award Number R01 DK124145 (Thorndike) and T32DK007028 (Gu). The content is solely the responsibility of the authors and does not necessarily represent the official views of the National Institutes of Health.”

3. In the online submission form you indicate that your data is not available for proprietary reasons and have provided a contact point for accessing this data. Please note that your current contact point is a co-author on this manuscript. According to our Data Policy, the contact point must not be an author on the manuscript and must be an institutional contact, ideally not an individual. Please revise your data statement to a non-author institutional point of contact, such as a data access or ethics committee, and send this to us via return email. Please also include contact information for the third party organization, and please include the full citation of where the data can be found.

Additional Editor Comments:

Please address the reviewer's comments.

Reviewers' comments:

Reviewer's Responses to Questions

**Comments to the Author**

1. Is the manuscript technically sound, and do the data support the conclusions?

Reviewer #1: Yes

2. Has the statistical analysis been performed appropriately and rigorously? 

Reviewer #1: Yes

3. Have the authors made all data underlying the findings in their manuscript fully available?

Reviewer #1: Yes

4. Is the manuscript presented in an intelligible fashion and written in standard English?

Reviewer #1: Yes

5. Review Comments to the Author

Reviewer #1: Summary

In this study, authors used data from LiveWell observational cohort to identify the most important sociodemographic and behavioural factors related to the diet quality of low-income adults. Diet was assessed using 2x self-administered 24-h dietary recalls (ASA24) and the HEI-2020. Authors found that the most important factors associated with lower HEI-2020 were house ownership, male, tobacco use, marijuana use, and skipping meals. Strength of this study includes dietary assessment instrument and explicit/clear description of the HEI-2020 score calculation (ie, population ratio and simple scoring), comprehensive data collection and consideration of participants from hardly reached communities. I congratulate authors for their work. I only have minor comments.

Minor

The modelling approach and the underlying data are not adequate to confirm causal relationships, which is fine in the context of this descriptive study and adequately mentioned in limitations (L309). However, readers may be tempted to interpret findings as if acting on any factors would necessarily lead to health/diet improvements. I suggest authors further add a statement about this issue rather than only in limitations. In other words, I think it is relevant to remind readers how findings should be (correctly) interpreted early in the discussion.

L168. Please indicate how many bootstrap samples were used and the specific method (i.e., normal approximation or percentile bootstrap)?

L187-190. Proportion of missing data: to be consistent with the STROBE reporting checklist item 14 (Vandenbroucke et al., 2007), please add a table that indicates the number of participants with missing data for each variable of interest.

Interpretation (ie, L250). An HEI score of 63 is higher than mean scores typically reported in NHANES. For example, 56.6 was the mean score reported by (Reedy et al., 2018). I acknowledge there is room for improvement, but it does not seem entirely accurate to state that diet quality is *not* aligned with guidelines.

Code sharing. I recommend authors make the analytic code available in supplemental material or in a code repository.

References

Reedy, J., Lerman, J. L., Krebs-Smith, S. M., Kirkpatrick, S. I., Pannucci, T. E., Wilson, M. M., Subar, A. F., Kahle, L. L., & Tooze, J. A. (2018). Evaluation of the Healthy Eating Index-2015. J Acad Nutr Diet, 118(9), 1622-1633. https://doi.org/10.1016/j.jand.2018.05.019

Vandenbroucke, J. P., von Elm, E., Altman, D. G., Gotzsche, P. C., Mulrow, C. D., Pocock, S. J., Poole, C., Schlesselman, J. J., Egger, M., & Initiative, S. (2007). Strengthening the Reporting of Observational Studies in Epidemiology (STROBE): explanation and elaboration. PLoS Med, 4(10), e297. https://doi.org/10.1371/journal.pmed.0040297

6. PLOS authors have the option to publish the peer review history of their article (what does this mean?). If published, this will include your full peer review and any attached files.

Reviewer #1: No

---

## [Author Response · Author response to Decision Letter 0]

25 Jul 2024

Response: The files are now formatted and named as per journal requirements,

“Research reported in this publication was supported by the National Heart, Lung, and Blood Institute of the National Institutes of Health under Award Number T32HL098048 (Cheng) and K24 HL163073 (Thorndike) and the National Institute of Diabetes and Digestive and Kidney Diseases under Award Number R01 DK124145 (Thorndike) and T32DK007028 (Gu). The content is solely the responsibility of the authors and does not necessarily represent the official views of the National Institutes of Health.”

Response: Please use the following updated statement: Research reported in this publication was supported by the National Heart, Lung, and Blood Institute of the National Institutes of Health under Award Number T32HL098048 (Cheng) and K24 HL163073 (Thorndike) and the National Institute of Diabetes and Digestive and Kidney Diseases under Award Number R01 DK124145 (Thorndike) and T32DK007028 (Gu). The content is solely the responsibility of the authors and does not necessarily represent the official views of the National Institutes of Health. The funders had no role in study design, data collection and analysis, decision to publish, or preparation of the manuscript.

3. In the online submission form you indicate that your data is not available for proprietary reasons and have provided a contact point for accessing this data. Please note that your current contact point is a co-author on this manuscript. According to our Data Policy, the contact point must not be an author on the manuscript and must be an institutional contact, ideally not an individual. Please revise your data statement to a non-author institutional point of contact, such as a data access or ethics committee, and send this to us via return email. Please also include contact information for the third party organization, and please include the full citation of where the data can be found.

Response: Please contact the Massachusetts General Hospital Institutional Review board via email at partnersirb@partners.org to access data.

Response: We have reviewed all references for accuracy and have made no changes. 

Additional Editor Comments:

Please address the reviewer's comments.

Response: We thank the reviewer for their comments. Below, we respond to each comment. 

Comments to the Author

Reviewer #1: Summary

In this study, authors used data from LiveWell observational cohort to identify the most important sociodemographic and behavioural factors related to the diet quality of low-income adults. Diet was assessed using 2x self-administered 24-h dietary recalls (ASA24) and the HEI-2020. Authors found that the most important factors associated with lower HEI-2020 were house ownership, male, tobacco use, marijuana use, and skipping meals. Strength of this study includes dietary assessment instrument and explicit/clear description of the HEI-2020 score calculation (ie, population ratio and simple scoring), comprehensive data collection and consideration of participants from hardly reached communities. I congratulate authors for their work. I only have minor comments.

Minor

The modelling approach and the underlying data are not adequate to confirm causal relationships, which is fine in the context of this descriptive study and adequately mentioned in limitations (L309). However, readers may be tempted to interpret findings as if acting on any factors would necessarily lead to health/diet improvements. I suggest authors further add a statement about this issue rather than only in limitations. In other words, I think it is relevant to remind readers how findings should be (correctly) interpreted early in the discussion.

Response: We have added the following sentence to the beginning of the discussion section: “We caution that the associations between factors and diet quality cannot be interpreted causally; therefore, addressing any individual factor may not necessarily lead to direct improvement in diet quality.”

L168. Please indicate how many bootstrap samples were used and the specific method (i.e., normal approximation or percentile bootstrap)?

Response: We updated the description of bootstrap resampling in the analysis section as such: “Bootstrap resampling with 200 bootstraps was utilized to estimate 95% confidence intervals (CIs) using the percentile method as appropriate since the bootstrap distribution was approximately normal.”

L187-190. Proportion of missing data: to be consistent with the STROBE reporting checklist item 14 (Vandenbroucke et al., 2007), please add a table that indicates the number of participants with missing data for each variable of interest.

Response: We have added a supplemental table, Supplemental Table 1, that indicates the number of participants with missing data for factors included in the LASSO model. In the analysis section, when discussing imputation of missing data, we refer readers to this supplemental table. 

Interpretation (ie, L250). An HEI score of 63 is higher than mean scores typically reported in NHANES. For example, 56.6 was the mean score reported by (Reedy et al., 2018). I acknowledge there is room for improvement, but it does not seem entirely accurate to state that diet quality is *not* aligned with guidelines.

Response: Instead of stating that the HEI score of 63 was “not aligned” with dietary guidelines, as the reviewer suggests, we have moderated the wording. Instead, we state that the HEI score “could be improved to better align with dietary guidelines.” We believe this statement is appropriate given that the score for our sample is below the score of 74 that would meet Healthy People 2020 goals (see citation 41: Wilson, 2016). We reference this citation in the paragraph that immediately follows the revised statement.

Code sharing. I recommend authors make the analytic code available in supplemental material or in a code repository.

Response: We would be happy to share analytical code. In the Data Availability Statement, we wish to indicate information about the access of the analytic code: “Analytical code used in this analysis to generate results can be accessed via Open Science Framework (DOI 10.17605/OSF.IO/WY4XF). Citation: Cheng J. Analytic Code: Sociodemographic and Behavioral Factors Associated with Diet Quality among Low-income Community Health Center Patients with Hypertension. doi:10.17605/OSF.IO/WY4XF.”

---

## [Decision Letter · Decision Letter 1]

15 Aug 2024

PONE-D-24-06116R1Sociodemographic and Behavioral Factors Associated with Diet Quality among Low-income Community Health Center Patients with HypertensionPLOS ONE

Dear Dr. Cheng,

Thank you for submitting your manuscript to PLOS ONE. After careful consideration, we feel that it has merit but does not fully meet PLOS ONE’s publication criteria as it currently stands. Therefore, we invite you to submit a revised version of the manuscript that addresses the points raised during the review process.

We look forward to receiving your revised manuscript.

Kind regards,

Jinyi Wu, MD

Academic Editor

PLOS ONE

Journal Requirements:

**Comments to the Author**

1. If the authors have adequately addressed your comments raised in a previous round of review and you feel that this manuscript is now acceptable for publication, you may indicate that here to bypass the “Comments to the Author” section, enter your conflict of interest statement in the “Confidential to Editor” section, and submit your "Accept" recommendation.

Reviewer #1: All comments have been addressed

Reviewer #2: (No Response)

2. Is the manuscript technically sound, and do the data support the conclusions?

Reviewer #1: Yes

Reviewer #2: Partly

3. Has the statistical analysis been performed appropriately and rigorously? 

Reviewer #1: Yes

Reviewer #2: Yes

4. Have the authors made all data underlying the findings in their manuscript fully available?

Reviewer #1: Yes

Reviewer #2: Yes

5. Is the manuscript presented in an intelligible fashion and written in standard English?

Reviewer #1: Yes

Reviewer #2: Yes

6. Review Comments to the Author

Reviewer #1: I thank the authors for diligently addressing my previous comments. Thank you for this work. I have no additional feedback to provide.

Reviewer #2: In this study, the authors used data from a low-income, urban cohort to identify correlates of diet quality among individuals with hypertension. Interventions to improve the health and well-being of underserved populations is important, however, I have some serious concerns about the paper in its present form. This seems like some of the exploratory analysis that would be necessary in informing intervention design, however, I do not think there is enough here for a standalone paper. I am not sure that it adds much to the literature beyond what is already known. I am also concerned about the discussion of certain factors as causal or as a point for intervention (e.g., smoking behavior). While smoking and diet are correlated, there is no causal relationship, and while the authors acknowledge this, they suggest using smoking behavior to inform their intervention anyways. This also has the potential to reinforce stigmas against those the intervention seeks to serve, and so I recommend that the authors consider reframing the discussion around their findings.

Additional comments:

The authors mention that NIH/AHA recommend CHW-led interventions (refs. 11,12). Are there any examples of successful CHW-led interventions to improve dietary quality among low-income populations? If so it may be helpful to reference these; if not it may be helpful to explicitly mention that this is a gap that the study team aims to fill.

While I appreciate that this is a secondary data analysis, it seems like the best way to determine what sorts of interventions would work well for this population would be to ask the participants directly. I recommend some qualitative work to support the design of this intervention. It would also help contextualize some of the results reported here, such as meal skipping. The reason for meal skipping (income, time constraints, lack of access to appropriate food preparation facilities) is likely important for the participant’s overall diet quality and for the success of the intervention. If qualitative work is planned, I recommend mentioning this explicitly as something that will support the overall intervention design.

Line 60 - A bit confusing why CHWs would seek to prioritize certain factors over the others – particularly for sociodemographic behaviors, but also for behavioral factors as these are correlates but not causal. I recommend cutting this paragraph.

Line 79. Much of this data collection took place during the first months of the COVID pandemic. This was a time of unprecedented disruption to people’s lives, the food system, and the healthcare system. There were massive surges in food insecurity in the US during these months. I suggest the authors add a sentence in the limitations acknowledging the potential role of COVID on the data collection.

Lines 95-96 Please clarify what is meant by “Gender, race, education, marital status, and employment categories were collapsed because of small cell sizes.” It would be helpful if there was a supplemental table that detailed what all categorical options were and how categories were collapsed.

Lines 110-120 are their full lists of what is included in these scales? And are there citations for these measures? Many of the other variables included come from validated questionnaires with citations (e.g., General Anxiety Disorders Questionnaire). It would be helpful if the full response options were provided in a supplement, and if the validations of these measures were cited (where available).

Lin 154. Why was HEI the metric of diet quality used and not (for example) DASH, which is mentioned in both the introduction and the discussion as a dietary pattern to address hypertension, or A-HEI, which has stronger associations with BP? I suggest the authors add a sentence justifying the use of HEI as the indicator of diet quality rather than one of these other metrics.

The discussion could benefit from discussion of different interventions and policies beyond CHWs to improve diet and hypertension among this population. For example, we know that cash transfers are more impactful in improving diet and health, especially among low-income populations such as that included in this study.

The authors mention time constraints as a barrier to healthy diet among low-income interventions. How would a CHW-led intervention help to address time constraints experienced by these individuals? Wouldn’t participation in such an intervention just be an additional time pressure?

Figure 2 – this is a bit hard to read as constructed, I recommend adding additional spacing between the different factors to make it easier to read

7. PLOS authors have the option to publish the peer review history of their article (what does this mean?). If published, this will include your full peer review and any attached files.

Reviewer #1: No

Reviewer #2: No

---

## [Author Response · Author response to Decision Letter 1]

16 Aug 2024

Response to Reviewers

Reviewer #2: In this study, the authors used data from a low-income, urban cohort to identify correlates of diet quality among individuals with hypertension. Interventions to improve the health and well-being of underserved populations is important, however, I have some serious concerns about the paper in its present form. This seems like some of the exploratory analysis that would be necessary in informing intervention design, however, I do not think there is enough here for a standalone paper. I am not sure that it adds much to the literature beyond what is already known. I am also concerned about the discussion of certain factors as causal or as a point for intervention (e.g., smoking behavior). While smoking and diet are correlated, there is no causal relationship, and while the authors acknowledge this, they suggest using smoking behavior to inform their intervention anyways. This also has the potential to reinforce stigmas against those the intervention seeks to serve, and so I recommend that the authors consider reframing the discussion around their findings.

Response: Thank you for this thoughtful suggestion. We have removed any sentences which may have indicated causal relationships between social factors (e.g., smoking) and dietary intake or reinforced stigma (e.g., we have cut the sentence: “Awareness of substance use patterns could help contextualize dietary guidance provided by a CHW and provide an opportunity for discussing the relationship of the substance, such as tobacco or alcohol use, with hypertension.”. 

Additional comments:

The authors mention that NIH/AHA recommend CHW-led interventions (refs. 11,12). Are there any examples of successful CHW-led interventions to improve dietary quality among low-income populations? If so it may be helpful to reference these; if not it may be helpful to explicitly mention that this is a gap that the study team aims to fill.

Response: This is a helpful suggestion. We note the similarities between our current and planned work as well as how our work expands on that which has already been done: “CHW-delivered hypertension programs are promising ways of reaching diverse participants in consultation with, and outside of, primary care (44, 45). Other CHW-led interventions with dietary components show potential effectiveness in reducing blood pressure and improving health behaviors although diet quality measures, such as the HEI, are not evaluated (46-48). As with other CHW-led hypertension interventions, our planned intervention development and evaluation will include qualitative work to complement our quantitative analyses (46, 49). Thus, we have planned semi-structured interviews with both patients and CHWs, in order to contextualize our current findings and to explore if and how to address these factors in our intervention design and delivery (e.g., involving female partners in the intervention). Thus, our work utilizes similar practices as other promising interventions and expands on existing literature as we will explicitly focus on robust measurement of diet quality outcomes.

While I appreciate that this is a secondary data analysis, it seems like the best way to determine what sorts of interventions would work well for this population would be to ask the participants directly. I recommend some qualitative work to support the design of this intervention. It would also help contextualize some of the results reported here, such as meal skipping. The reason for meal skipping (income, time constraints, lack of access to appropriate food preparation facilities) is likely important for the participant’s overall diet quality and for the success of the intervention. If qualitative work is planned, I recommend mentioning this explicitly as something that will support the overall intervention design.

Response: We completely agree with the reviewer’s comment, and we are in the process of conducting qualitative work to build on and complement this analysis. As recommended by the reviewer we add a sentence explicitly acknowledging the value of this qualitative work: “As with other CHW-led hypertension interventions, our quantitative analysis should be complemented by qualitative work. Thus, we have planned semi-structured interviews with both patients and CHWs, in order to contextualize our current findings and to explore if and how to address these factors in our intervention design and delivery (e.g., involving female partners in the intervention).”

Line 60 - A bit confusing why CHWs would seek to prioritize certain factors over the others – particularly for sociodemographic behaviors, but also for behavioral factors as these are correlates but not causal. I recommend cutting this paragraph.

Response: We have cut this paragraph as recommended.

Line 79. Much of this data collection took place during the first months of the COVID pandemic. This was a time of unprecedented disruption to people’s lives, the food system, and the healthcare system. There were massive surges in food insecurity in the US during these months. I suggest the authors add a sentence in the limitations acknowledging the potential role of COVID on the data collection.

Response: Thank you. We have added a sentence to this effect: “Finally, this was a relatively small sample of patients from one urban health system in the northeast US and data collection occurred during the COVID-19 pandemic. Thus, results may not be generalizable to patients from rural areas or from other regions within the US or outside the US, and the pandemic might have temporarily increased the reporting of social needs and/or lead to the worsening of diet quality.”

Lines 95-96 Please clarify what is meant by “Gender, race, education, marital status, and employment categories were collapsed because of small cell sizes.” It would be helpful if there was a supplemental table that detailed what all categorical options were and how categories were collapsed.

Response: We have provided analytical code (and formats) on open science framework (DOI 10.17605/OSF.IO/WY4XF) which includes information on the categories before and after collapsing.

Lines 110-120 are their full lists of what is included in these scales? And are there citations for these measures? Many of the other variables included come from validated questionnaires with citations (e.g., General Anxiety Disorders Questionnaire). It would be helpful if the full response options were provided in a supplement, and if the validations of these measures were cited (where available).

Response: For the 3-item housing related questions we have added two citations about the development and validation. The other items in this paragraph were created to be relevant to our population and resources. These items on living site problems, receiving housing assistance, receiving transportation assistance, and help obtaining these types of assistance have been added to a supplement. We direct readers in this paragraph to view the supplement. We also note that our analytical code which is available on open science framework (DOI 10.17605/OSF.IO/WY4XF) does provide descriptions of each of these items (i.e., a variable label) as well as for all items included in the analysis. We hope that this is sufficient detail for the understanding of readers. 

Lin 154. Why was HEI the metric of diet quality used and not (for example) DASH, which is mentioned in both the introduction and the discussion as a dietary pattern to address hypertension, or A-HEI, which has stronger associations with BP? I suggest the authors add a sentence justifying the use of HEI as the indicator of diet quality rather than one of these other metrics.

Response: Thank you. We have added a sentence indicating that the HEI was used as it aligns with the US Dietary Guidelines for Americans. While preferences may differ on which index to use, we find it useful in our work to align with the guidelines as they are familiar to a broad segment of both the research community and the general population: “We chose to use the Healthy Eating Index-2020 (HEI-2020) (28, 29) as the measure of diet quality in this study as it is a valid and reliable measure of overall diet quality aligned with the Dietary Guidelines for Americans.”

We have also added a sentence showing that the HEI does relate to blood pressure and is correlated with other indices, specifically DASH and AHEI: “The HEI has been shown to relate to blood pressure, (30, 31) correlate with other diet quality indices, and shows a similar relationship to CVD mortality as other indices (32).” 

The discussion could benefit from discussion of different interventions and policies beyond CHWs to improve diet and hypertension among this population. For example, we know that cash transfers are more impactful in improving diet and health, especially among low-income populations such as that included in this study.

Response: As the reviewer indicates, the purpose of this analysis was to inform a CHW healthy eating intervention as it is being developed through both quantitative and qualitative means. We believe that additional discussion of the effectiveness of non-CHW nutrition education forms of intervention such as cash transfer programs is beyond the scope of this paper. However, we do agree with the reviewer that cash transfers are a very promising form of intervention for low socioeconomic groups that should be further researched and promoted. 

To address the role of financial support or incentives as a potential strategy to improve diet, we have added to our discussion: “…connecting patients with community-based nutrition services may help improve the diet of individuals with hypertension as our model suggested that receiving support for food acquisition (i.e., meals at a community organization and receiving a supermarket gift card) was positively associated with diet quality in this mostly low-income population.”

The authors mention time constraints as a barrier to healthy diet among low-income interventions. How would a CHW-led intervention help to address time constraints experienced by these individuals? Wouldn’t participation in such an intervention just be an additional time pressure?

Response: Thank you for this comment. Although participating in the CHW program requires time, the goal is to make the intervention as convenient and efficient as possible for the participant. For example, the CHW will provide guidance by phone or video call at a time and place that works for the participant, and therefore this does not require time spent traveling to the clinic for a visit with a dietitian or other clinician. 

In addition, our goal is that through participating in the CHW program, participants can learn how to address other time constraints specific to shopping and food preparation and that they can use these skills during and after the program has ended. We have added the sentence: “For example…meal planning, and quick healthy go-to meals. We note that while participation in a phone- and web-based CHW -led program is a temporary time constraint while enrolled, we hope that CHWs can help patients develop and/or strengthen skills that reduce the time burden associated with shopping and cooking.” 

Figure 2 – this is a bit hard to read as constructed, I recommend adding additional spacing between the different factors to make it easier to read

Response: Thank you for this suggestion. We have increased spacing.

---

## [Decision Letter · Decision Letter 2]

6 Sep 2024

Sociodemographic and Behavioral Factors Associated with Diet Quality among Low-income Community Health Center Patients with Hypertension

PONE-D-24-06116R2

Dear Dr. Cheng,

We’re pleased to inform you that your manuscript has been judged scientifically suitable for publication and will be formally accepted for publication once it meets all outstanding technical requirements.

Kind regards,

Jinyi Wu, MD

Academic Editor

PLOS ONE

Additional Editor Comments (optional):

Reviewers' comments:

Reviewer's Responses to Questions

**Comments to the Author**

1. If the authors have adequately addressed your comments raised in a previous round of review and you feel that this manuscript is now acceptable for publication, you may indicate that here to bypass the “Comments to the Author” section, enter your conflict of interest statement in the “Confidential to Editor” section, and submit your "Accept" recommendation.

Reviewer #1: All comments have been addressed

Reviewer #3: All comments have been addressed

2. Is the manuscript technically sound, and do the data support the conclusions?

Reviewer #1: Yes

Reviewer #3: Yes

3. Has the statistical analysis been performed appropriately and rigorously? 

Reviewer #1: Yes

Reviewer #3: Yes

4. Have the authors made all data underlying the findings in their manuscript fully available?

Reviewer #1: Yes

Reviewer #3: Yes

5. Is the manuscript presented in an intelligible fashion and written in standard English?

Reviewer #1: Yes

Reviewer #3: Yes

6. Review Comments to the Author

Reviewer #1: (No Response)

Reviewer #3: (No Response)

7. PLOS authors have the option to publish the peer review history of their article (what does this mean?). If published, this will include your full peer review and any attached files.

Reviewer #1: No

Reviewer #3: No

---

## [Editor Report · Acceptance letter]

10 Sep 2024

PONE-D-24-06116R2 

PLOS ONE

Dear Dr. Cheng, 

I'm pleased to inform you that your manuscript has been deemed suitable for publication in PLOS ONE. Congratulations! Your manuscript is now being handed over to our production team.

Kind regards, 

on behalf of

Dr. Jinyi Wu 

Academic Editor

PLOS ONE